

# Nonlinear effects in 4D-Var

Massimo Bonavita[1], Peter Lean[1], Elias Holm[1]

[1]European Centre for Medium-Range Weather Forecasts, Shinfield Park, Reading, RG2 9AX, UK

*Correspondence to*: Massimo Bonavita (massimo.bonavita@ecmwf.int)

**Abstract.** The ability of a data assimilation system to deal effectively with nonlinearities arising from the prognostic model or the relationship between the control variables and the available observations has received a lot of attention in theoretical studies based on very simplified test models. Less work has been done to quantify the importance of nonlinearities in operational, state-of-the-art global data assimilation systems. In this paper we analyse the nonlinear effects present in ECMWF 4D-Var and evaluate the ability of the incremental formulation to solve the nonlinear assimilation problem in a realistic NWP environment.

We find that nonlinearities have increased over the years due to a combination of increased model resolution and the ever-growing importance of observations that are nonlinearly related to the state. Incremental 4D-Var is well suited for dealing with these nonlinear effects, but at the cost of increasing the number of outer loop relinearisations. We then discuss strategies for accommodating the increasing number of sequential outer loops in the tight schedules of operational global NWP.

## 1 Introduction

The importance of nonlinear effects has been recognised since the early days of the development of 4D-Var (e.g., Gauthier, 1992; Miller et al., 1994; Pires et al., 1996). The presence of nonlinearities in either the model or the observations can potentially cause significant deviations from the usual Gaussian distribution assumed to describe observation and background errors in the definition of the 4D-Var cost function. This in turn translates into a more complex topology of the cost function and the potential for multiple minima (e.g., Pires et al., 1996; Hoteit, 2008). In these conditions, finding the global minimum

of the 4D-Var cost function for realistic numerical weather prediction (NWP) applications becomes computationally unaffordable and, even if it were possible, the interpretation and usefulness of the result in the case of multi-modal error distributions becomes unclear in a deterministic analysis context (Lorenc and Payne, 2007).

In order to make the variational problem computationally tractable and mathematically well-posed, simplifications are required. In particular, the use of linear model and observation operators leads to strictly quadratic cost functions, which brings

two major benefits: a) it guarantees the convergence of the minimisation algorithm to the global minimum and b) it allows the use of efficient, gradient-based iterative minimisation algorithms (Fisher, 1998). This consideration spurred research in NWP applications of variational methods towards perturbative solution algorithms, where the full nonlinear minimization problem is approximated as a series of quadratic cost functions obtained by repeated linearisations around progressively more accurate guess values of the solution. This idea, based on the general Gauss-Newton method for the solution of nonlinear least squares





problems (Björck, 1996), was first introduced in the meteorological literature by Courtier, Thépaut and Hollingsworth, 1994 (CTH in the following) as "Incremental 4D-Var". In that paper the main stated objective of incremental 4D-Var was the reduction of the computational costs of full 4D-Var in order to make it feasible for operational application. Its ability to deal with weak nonlinearities was also noted and subsequently investigated in simplified models, particularly in relation to the

length of the assimilation window and the global convergence properties of the algorithm (e.g., Tanguay et al., 1994; Laroche and Gauthier, 1998). After the operational implementation of incremental 4D-Var at ECMWF (Rabier et al., 2000) and, later, in other major global NWP Centres (Kadowaki, 2005; Rosmond and Xu, 2006; Gauthier et al., 2007; Rawlins et al., 2007) the possibility arose to address in realistic NWP settings still open questions about the limits of applicability of 4D-Var in nonlinear situations. A series of studies (Andersson et al., 2005; Radnòti et al., 2005; Trémolet, 2004, 2007) conducted with the ECMWF

Integrated Forecasting System (IFS) provided answers to some of these questions in the context of the ECMWF operational system of the time. These studies emphasised the importance of the consistency between the nonlinear and linearised evolution of the analysis increments during the assimilation window for the global convergence of the incremental 4D-Var. This, in turn, was shown to require the availability of accurate linearised models, and the need to run inner and outer loops with not too large discrepancies in terms of spatial resolution and time step (a ratio of three between the outer and inner loop resolutions was

found to give satisfactory results). As there is no guarantee of global convergence of the incremental 4D-Var algorithm, the aforementioned studies also stressed the importance of regularly re-evaluating the nonlinearity issues in future operational systems. From the time of these investigations, the operational ECMWF IFS has changed considerably. From the perspective of the validity of the linearity assumptions in the incremental formulation, two changes are particularly relevant: a) the increase in resolution at both outer loop and inner loop level and b) the introduction of a very large number of humidity, cloud and

precipitation-sensitive satellite observations in the analysis system (Geer et al., 2017). In terms of spatial resolution, the effective grid spacing has gone from approx. 40 Km (TL511) to approx. 9 Km (TCo1279), for the 4D-Var outer loops, and from approx. 130 Km (TL159) to approx. 50 Km (TL399) for the inner loops. Thus, nonlinearities are expected to play a larger role in the current IFS, also in view of the fact that the ratio between the resolutions of the outer and inner loops of the minimization has increased from approx. 3.2 to 5.5. In terms of observation usage, the increase in the number and influence

of humidity, cloud and precipitation-sensitive observations can also be expected to expose nonlinear effects connected to the way their observation operators respond to forecasted humidity and precipitation structures. Some of these issues were already described at the time of the introduction of the "all-sky" framework for the assimilation of microwave imagers sensitive to humidity and precipitation (Bauer et al., 2010), but at that time the number and influence of these observation types on the 4D-Var analysis was relatively small. Currently, however, all-sky observations are one of the most important components of the

observing system used operationally at ECMWF (Geer et al., 2017) and it is thus important to understand the capabilities and limitations of 4D-Var to deal with this type of nonlinearities.

This paper is organised as follows. In Sec. 2, we briefly review the incremental 4D-Var algorithm in order to highlight the hypotheses underlying the tangent linear approximation and the mathematical basis of the outer loop iterations. In Sec. 3 evidence of nonlinear effects in current ECMWF 4D-Var is presented, from both an observational and a model perspective. In





Sec. 4, we evaluate how effective incremental 4D-Var is in dealing with both observation and model nonlinearities. Sec. 5 addresses the question of how important the ability to run outer loops is in the current ECMWF data assimilation system, in terms of both analysis and forecast skill. These results and their implication for data assimilation strategy at ECMWF and elsewhere are discussed in Sec. 6.

## 2 Algorithmic Aspects

The aim of variational data assimilation is to determine the model trajectory that best fits in a least squares sense the observations available during a given time window. This concept naturally leads to the formulation of the standard strong constraint 4D-Var cost function:

$$J(\mathbf{x}_0) = \frac{1}{2}(\mathbf{x}_b - \mathbf{x}_0)^{\mathrm{T}}\mathbf{B}^{-1}(\mathbf{x}_b - \mathbf{x}_0) + \frac{1}{2}\sum_{k=0}^{K}\big(\mathbf{y}_k - G_k(\mathbf{x}_0)\big)^{\mathrm{T}}\mathbf{R}_k^{-1}\big(\mathbf{y}_k - G_k(\mathbf{x}_0)\big) = J_{\mathbf{B}}(\mathbf{x}_0) + J_{\mathbf{O}}(\mathbf{x}_0) \qquad (1)$$

In (1) $\mathbf{x}_0$ is the control vector at the start of the assimilation window; $\mathbf{x}_b$ and $\mathbf{B}$ are the background and its expected error covariance matrix; $\mathbf{y}_k$ and $\mathbf{R}_k$ are the set of observations presented to the analysis in the k sub-window and their expected error covariances; and $G_k$ is a generalised observation operator (or forward model) that produces the model equivalents of the observations $\mathbf{y}_k$ by first integrating the prognostic model from $t_0$ to $t_k$ and then applying the standard observation operator $\mathbf{H}_k$ to the propagated fields, i.e.:

$\quad G_k = H_k \circ M_{t_0 \rightarrow t_k}$ $\qquad$ (2)

The formulation (1) represents the general nonlinear weighted least square solution of the assimilation problem using the forecast model as a strong constraint. Problem (1) cannot however be solved efficiently by standard optimal control methods for realistic numerical weather prediction (NWP) data assimilation systems, given the size of the control vector $\mathbf{x}_0$ (O($10^9$)). A possible solution, first proposed in CTH, 1994, under the name of "Incremental 4D-Var", is to simplify the solution of (1)

through the application of an approximated form of the Gauss-Newton method (Lawless et al., 2005; Gratton et al., 2007). This consists of approximating the minimization of the nonlinear cost function (1) as a sequence of minimizations of linear, quadratic cost functions defined in terms of perturbations around a sequence of progressively more accurate trajectories (i.e., nonlinear model integrations). The cost function linearised around a trajectory $\mathbf{x}^t$ can be expressed as an exact quadratic problem in terms of the increment at the initial time $\delta\mathbf{x}_0$:

$\quad J(\delta\mathbf{x}_0) = \frac{1}{2}(\delta\mathbf{x}_0 + \mathbf{x}_0^t - \mathbf{x}_b)^{\mathrm{T}}\mathbf{B}^{-1}(\delta\mathbf{x}_0 + \mathbf{x}_0^t - \mathbf{x}_b) + \frac{1}{2}\sum_{k=0}^{K}\big(\mathbf{d}_k - \mathbf{G}_k(\delta\mathbf{x}_0)\big)^{\mathrm{T}}\mathbf{R}_k^{-1}\big(\mathbf{d}_k - \mathbf{G}_k(\delta\mathbf{x}_0)\big)$ $\qquad$ (3)

In Eq. (3) $\mathbf{d}_k = \mathbf{y}_k - G_k(\mathbf{x}_0^t)$ are the observation departures around the latest model trajectory and $\mathbf{G}_k = \mathbf{H}_k \mathbf{M}_{t_0 \rightarrow t_k}$ is the linearisation of the generalised observation operator around the defined trajectory.

In the observation part of the cost function, the so-called "tangent linear (TL) approximation" has been made in going from (1) to (3):

$\quad \mathbf{y}_k - G_k(\mathbf{x}_0) = \mathbf{y}_k - G_k(\mathbf{x}_0^t + \delta\mathbf{x}_0) =$


$$= \mathbf{y}_k - G_k(\mathbf{x}_0^t) - \mathbf{G}_k(\delta\mathbf{x}_0) - \frac{1}{2}(\delta\mathbf{x}_0)^{\mathrm{T}}\left(\frac{\partial\mathbf{G}_k}{\partial\mathbf{x}}\right)_{\mathbf{x}^t}(\delta\mathbf{x}_0) - O(\|\delta\mathbf{x}_0\|^3) \approx \mathbf{y}_k - G_k(\mathbf{x}_0^t) - \mathbf{G}_k(\delta\mathbf{x}_0) \qquad (4)$$

In the Taylor expansion in Eq. (4), terms of $O(\|\delta\mathbf{x}_0\|^2)$ and higher are neglected (note that if (4) is exactly satisfied, then Eq. (3) is equivalent to Eq. (1)). This approximation, as first noted in Lawless et al, 2005, is equivalent to the standard approximation used in the Gauss-Newton optimization algorithm, i.e. neglecting the second order derivatives of $G_k$ in the Hessian of the cost function:

$$\nabla^2 J = \mathbf{B}^{-1} + \sum_{k=0}^{K}\left(\mathbf{G}_k(\delta\mathbf{x}_0)\right)^{\mathrm{T}}\mathbf{R}_k^{-1}\left(\mathbf{G}_k(\delta\mathbf{x}_0)\right) - \sum_{k=0}^{K}\left(\frac{\partial\mathbf{G}_k}{\partial\mathbf{x}}\right)_{\mathbf{x}^t}\mathbf{R}_k^{-1}\left(\mathbf{y}_k - G_k(\mathbf{x}_0)\right) \approx$$

$$\mathbf{B}^{-1} + \sum_{k=0}^{K}\left(\mathbf{G}_k(\delta\mathbf{x}_0)\right)^{\mathrm{T}}\mathbf{R}_k^{-1}\left(\mathbf{G}_k(\delta\mathbf{x}_0)\right) \qquad (5)$$

The validity of the tangent linear approximation is thus based on whether either the increments $\delta\mathbf{x}_0$ are in some sense small or the dependence of the linearization of $G_k$ (i.e., $\mathbf{G}_k = \mathbf{H}_k\mathbf{M}_{t_0 \to t_k}$) to the reference trajectory is negligible. Concerning the first aspect, we only note here that the size of analysis increments is, to first order, a linear function of observation departures. Thus the size of departures need to be small with respect to the observation and background errors used in the analysis update for the TL approximation to hold (The interested reader can find further details in Bonavita et al., 2017b). The other aspect affecting the validity of the TL approximation relies on an implicit linearity assumption of both the forecast model and the observation operator in a neighbourhood of the reference trajectory. Experience at ECMWF indicates that there is a clear sensitivity of both the linearised observation operator and the linearised model to the linearisation state (e.g., Bauer et al., 2010; Janisková and Lopez, 2013). It is thus relevant to revisit the roles of model and observation nonlinearities in the current operational ECMWF 4D-Var implementation and validate the effectiveness of the incremental 4D-Var method in dealing with these nonlinearities.

Other, possibly less well-known, sources of nonlinearities in the ECMWF incremental 4D-Var formulation stems from the variational quality control (VarQC) of the observations and the nonlinear change of variable used for the humidity analysis. The VarQC algorithm is based on the Huber norm (Tavolato and Isaksen, 2015) and has the effect of making the observation error matrix $\mathbf{R}$ a function of the current departure $\mathbf{d}_k$ and thus of the reference state. However, as it is currently applied to conventional observation only, its impact on the linearity of the minimisation is limited. The other source of nonlinearity arises from the nonlinear change of variable used in the humidity analysis (Holm et al., 2003), which implies a nonlinear dependence of the $J_\mathbf{B}$ part of the cost function on the state. Consistently with the incremental 4D-Var philosophy, this is handled by a linear update of the humidity control variable in the quadratic cost function (3), followed by a nonlinear update of the humidity field at the outer loop level to provide the initial state for the new reference trajectory. The nonlinear effects connected with the humidity control variable are intimately linked with the usage of the all-sky observations, which provide the vast majority of humidity sensitive observations, and will be discussed in that context.



## 3 Evidence of nonlinear effects in 4D-Var

### 3.1 The role of the model

Model nonlinearities affect the 4D-Var solution in two main ways. First, the more nonlinear the high-resolution trajectory solution is, the spatially noisier the low-resolution interpolated linearization state for the 4D-Var inner loops becomes. This

roughness of the interpolated trajectory increases when differences between the timesteps and resolutions of the inner loops and the trajectory become larger. Second, the tangent linear evolution differs more from the nonlinear solution as nonlinearities increase. One measure of the degree of nonlinearity is to take the difference between the nonlinearly and linearly evolved increments in the last minimization:

$$M(x^{n-1} + \delta x^n) - (M(x^{n-1}) + \mathbf{M}\delta x^n) \tag{6}$$

and the globally averaged profile of the standard deviation of this quantity is shown in Fig. 1 for selected years from 2004 to 2017. Over the years, there has been an increase in the resolution of the trajectory and the inner loops and the gap in resolution between the two has increased. This has resulted in increased differences, which we interpret as increased nonlinearity due to the combination of increased model resolution and resolution differences between the inner loop and the trajectory. One way to counteract the nonlinearity that comes with resolution increases is to shorten the length of the assimilation window. This

can be achieved either by very short windows or by the use of overlapping assimilation windows. In this second case the reduction in nonlinearity is realised by reducing the size of the analysis increments $\delta x^n$ as each new window will start from a first guess trajectory that has already seen the observations in the overlapped part of the window.

### 3.2 The role of the observations

The significance of nonlinearities in the observation operators can be estimated using statistics from the Ensemble of Data

Assimilations (EDA, Isaksen et al., 2010) system which is run operationally at ECMWF. Each ensemble member is initialised using a perturbed model state with perturbations drawn from a distribution with zero mean. For linear observation operators, it follows that the ensemble mean of the model equivalents provided by the observation operators is expected to be close to the unperturbed control member (in fact, it should match it exactly in the limit of infinite ensemble size):

$$\frac{1}{N}\sum_{i=0}^{N} G_k(x_0 + \delta x_i) = \frac{1}{N}\sum_{i=0}^{N} G_k(x_0) + \frac{1}{N}\sum_{i=0}^{N} G_k(\delta x_i) = G_k(x_0) + G_k(\sum_{i=0}^{N} \delta x_i) \cong G_k(x_0) \tag{7}$$

Figures 2a-d show the relationship between the ensemble mean model equivalent value and the model equivalent values in the unperturbed control member for different observation types. For observations sensitive to tropospheric temperature, such as the Advanced Microwave Sounding Unit A (AMSU-A) channel 6 (Fig 2a) and radiosonde temperature observations (Fig 2b), a strong linear relationship holds, indicating that nonlinear effects in the observation operators are negligible. However, operators associated with observations sensitive to humidity and cloud show more significant nonlinear behaviour. For

example, the Advanced Technology Microwave Sounder (ATMS) channel 20 (Fig 2c) is sensitive to humidity and the Advanced Microwave Scanning Radiometer 2 (AMSR-2) channel 11 (Fig 2d) is sensitive to cloud liquid water.



## 4 Dealing with nonlinearities through the incremental approach

The incremental approach to 4D-Var (CTH, 1994) reduces the resolution of the inner loops to make the solution more affordable. Observation departures are calculated at high resolution and then the high-resolution trajectory is truncated and interpolated to the resolution of the inner loop for each timestep of the low-resolution minimization (Trémolet, 2004). At the

end of the minimization, the increments are projected back to the high resolution and added to the previous trajectory at the start of the assimilation window. This process is repeated for all minimizations, which can be at different resolutions, starting with the lowest resolution to capture the larger scales and increasing the resolution in later minimizations to extract more detailed information from the observations (Veerse and Thépaut, 1998).

### 4.1 Impact diagnostics in observation space

Bauer et al (2010) discussed how the difference in departures at the end of each minimisation step, and those in the subsequent nonlinear trajectory step (i.e., $\delta d_k = d_k^{non-linear} - d_k^{linear}$) indicate the presence of nonlinearities in the system. Figure 3 shows the standard deviation of the departures at each stage of 4D-Var for the AMSR-2 channel 10, which is sensitive to water vapour. It can be seen that each minimisation improves the fit between the model trajectory and the observations. However, the standard deviation in each nonlinear trajectory step is consistently higher than that at the end of the previous minimisation.

This is to be expected because of the resolution difference between nonlinear and linearised models and also due to the fact that nonlinear processes cannot be represented by the linear model and operators used in the minimisations.

Figure 4 plots the correlation coefficient and standard deviation of these differences at each outer loop, demonstrating that nonlinearities become smaller at each successive outer loop. For "linear" observation types such as radiosonde temperature and AMSU-A channel 6, the nonlinearities are less significant than for ATMS channel 20 (which is sensitive to humidity).

As expected, the departures for observations sensitive to cloud and humidity show increased nonlinear impacts. Figure 5 shows results from AMSR-2 channel 11 categorized using estimates of cloudiness from both the observations and the model fields (Geer and Bauer, 2011). It can be seen that the linear assumption holds less well for observations in cloudy regions compared to those in areas of clear sky.

### 4.2 Impact diagnostics in model space

A clear indicator of the success or otherwise of the incremental strategy is the size of the analysis increments produced by the linearised cost function (3) during successive outer loop iterations. For a well-behaved incremental 4D-Var converging towards the solution of the nonlinear cost function (1), successive analysis increments are expected to become smaller, reflecting the hypothesis that successive first guess trajectories provide increasingly accurate descriptions of the flow. This hypothesis is

supported by the experimental results shown in Fig. 6, where we present the vertical profiles of the standard deviations of the analysis increments of vorticity (left panel) and temperature (right panel) from a multi-incremental 4D-Var experiment with



five outer loops. The magnitude of the analysis increments is seen to gradually decrease for successive outer loop iterations, more rapidly in the stratosphere for vorticity. After five outer loop iterations, the magnitude of the analysis increments appears to be asymptote to a relatively small value for temperature throughout the atmospheric column ($\Delta T_a \approx 0.05K$), and for vorticity in the stratosphere and mesosphere ($\Delta vo_a \approx 10^{-7}s^{-1}$ for model levels greater than 70). On the other hand, incremental 4D-

Var does not seem to have fully converged for vorticity in the troposphere. This is confirmed by the longitudinal averages of the analysis increments produced by the first and the last outer loop for temperature, vorticity and humidity, which are shown in Fig. 7. It is apparent how the last outer loop iteration still manages to produce non-negligible increments for the tropospheric wind and humidity fields (middle and bottom rows in Fig. 7), as a result of the increased presence of nonlinear observations and the increased nonlinearity of the relevant meteorology (e.g., organised convection and baroclinic instability). This suggests

that increasing the number of outer loops from the current operational value of three up to at least five can lead to a better use of available observations and, ultimately, more accurate analyses and forecasts. An interesting side aspect of this investigation has been to highlight the relative large analysis increments produced by 4D-Var in the mesosphere (i.e., above model level 20 in the plots). This is due to a combination of relatively inaccurate model dynamics due to sponge layer effects and the scarcity of observational constraints in this part of the atmosphere (the highest peaking channels from current microwave sounders are

only marginally sensitive to this upper atmospheric layer).

### 4.3 A test case

An informative example of the effectiveness of incremental 4D-Var in dealing with nonlinear error evolution in active weather systems is described in the following test case of organised convection in the southern United States. These high-impact weather phenomena are particularly interesting from a data assimilation perspective because: 1) they have been shown to be

potential precursors of significant forecast "busts" in downstream regions, Europe in particular (Rodwell et al., 2013); and 2) they occur in probably the most densely observed region of the world, thus allowing a more in-depth look into the ability of the assimilation system to make effective use of the observations.

In the case described here, large scale organised convection with the satellite signature of a mesoscale convective complex (Fig. 8, left panel), was forming in the southern US coastal plains in the local evening hours of 2017-05-03, continuing for

most of the night. The synoptic situation, as depicted by the ECMWF operational analysis (Fig. 8, right panel) is characteristic of this type of events (Maddox, 1980): A strong warm, moist southerly flow from the Gulf of Mexico is taking place in the lower troposphere, in the region ahead of an upper level trough. The combination of strong warm and moist air advection in the lower levels with vorticity advection aloft leads to a situation conducive to intense organised convection in a region along the Texas-Louisiana cost, starting at around 13UTC on the 3 May 2017 and lasting until approx. 6UTC of the 4 May 2017.

Forecasting the intensity and location of convection is notoriously difficult and the ECMWF analysis increments (Fig. 9) show that the operational 4D-Var makes significant changes to the first guess fields throughout the atmospheric column. In particular, the analysis appears to adjust the strength of the convective system through a significant cooling at the top of the troposphere and associated enhancement of the divergent wind field (Figure 9, left panel). In the boundary layer (Figure 9, right panel), the



analysis increments show more spatial variability, but the main signal of localised warming and convergence of the wind field in the direction of movement of the convective system are apparent.

The magnitude of the analysis increments in the case studied here (up to ~8 K for temperature, ~30 m/s for wind) is more than an order of magnitude larger than their average standard deviations: thus, significant nonlinear effects are expected in the assimilation update. This is confirmed in Figure 10 where we show the standard deviation of the first guess and analysis departures from wind observations in the 100-400 hPa layer for the first guess and the one, three, five outer loop analyses over the 9 to 21 UTC assimilation window of 2017-05-03 (All experiments are performed at the operational TCo1279 resolution, approx. 9 km grid spacing, using IFS cycle 43R3). It is visually apparent that the analysed trajectories are better able to fit the observations with increasing number of outer loops: the area-averaged standard deviation of the innovations decreases from 2.301 m/s in the first guess trajectory to 2.117 m/s, 1.705 m/s and 1.622 m/s in the one, three and five outer loop analysis trajectories respectively. No further improvements were seen with further increases in the outer loop count, which points to residual model deficiencies, either in terms of spatial resolution and/or model errors. We note that the wind observations whose departures are shown in the plots in Fig. 10 come from the US radiosonde network and from aircraft observations, which implies that linear observation operators are used. Thus, the nonlinear effects seen in the plots arise exclusively from nonlinearities in the evolution of model perturbations in the assimilation window.

## 5 Results from cycling data assimilation experiments

The diagnostics presented in section 4 showed that increasing the number of outer loop iterations in the ECMWF 4D-Var helps to reduce the magnitude of nonlinearities in the analysis and suggests that it can lead to a better use of available observations, in particular those that are nonlinearly related to the model state. The next step is then to verify that these findings are confirmed in a cycled data assimilation environment as close as it is computationally affordable to the operational ECMWF assimilation system. To this end a series of data assimilation experiments has been run with a recent ECMWF IFS cycle (cycle 43R3, operational from July 2017), in which only the horizontal spatial resolution has been changed for both outer loops and inner loop minimizations. The operational 4D-Var runs three outer loops at TCo 1279 resolution (approx. 9 km) and performs three inner loop minimizations at TL255/TL319/TL399 resolution (approx. 80/60/50 km). In the experiments described here the outer loop resolution has been reduced to TCo399 (approx. 30 km) and the inner loop resolutions vary from TL95 to TL159 to TL255 (approx. 210, 125, 80 km; more details in Table 1). The number of outer loop updates varies from one to five. In the following, we present results for the one, three, four and five outer loop experiments. In these experiments, the full observational dataset used in operations at ECMWF has been assimilated. The number of inner loop iterations in the minimisations is not prescribed, because minimisations stop when a convergence criterion based on the information content of the minimisation is reached. Convergence is usually reached in approx. 30 iterations, and this number has been found not to be sensitive to resolution and number of outer loop relinearizations.



## 5.1 Analysis skill – full observing system

A standard way to evaluate the skill of the analyses produced by a cycling data assimilation system is to look at the statistics of observation minus analysis (o-a) departures and observation minus first guess departures (o-b). In the ECMWF 4D-Var the o-a departures are computed from a full model integration started from the analysed model state at the beginning of the 12
hour assimilation window. Thus, they give an indication of how closely a nonlinear forecast started from the initial analysis is able to fit observations throughout the assimilation window. The o-b departures are computed from a short-range forecast started from a 4D-Var analysis valid three hours before the start of the new assimilation window. Thus, the new observations are confronted with a nonlinear forecast in the 3 to 15 hour range. The o-b fit gives an indication of how much of the observation information from the previous assimilation window is retained in the short-range forecast used for cycling the analysis.

A representative sample of o-a and o-b departures is shown in Figure 11. In these plots the observation departures for a standard three outer loops 4D-Var assimilation cycle are used as a baseline against which the departures for one, four and five outer loop experiments are compared. The first thing to note is the significant degradation in both o-a and o-b statistics of the one-outer loop experiment. This degradation is visible for all observation types (not shown) and indicates that a linear analysis update is inadequate in the context of a 12-hour assimilation window. The other significant result is that increasing the number
of outer loops to four, and to a small extent five, can bring additional benefits in the tropospheric analysis, in particular for observations sensitive to humidity and clouds and for wind observations (not shown). This confirms the diagnostics of Figs. 6 and 7, i.e. that the wind and humidity analysis increments in the troposphere are still relatively large in the fourth and fifth outer loop updates, indicating that the minimization has not fully converged.

The only degradation in o-a and o-b statistics for the four and five outer loop experiments is visible in the stratospheric-peaking
channels of the microwave (Fig 11, bottom right, channels 10 to 14) and infrared hyperspectral instruments (not shown). This is particularly visible in the five outer loop experiment, where o-a statistics are clearly degraded for channels 10 to 14 (approx. peaking from 50 to 2 hPa) while the degradation in the fit to the short range forecast (o-b) is only marginal. This result can be partially explained by the diagnostic shown in Fig. 1, where it was shown that the magnitude of nonlinear effects in the ECMWF analysis system is relatively small in the stratosphere. Thus, the incremental minimisation can be expected to
converge more rapidly in the stratosphere and additional outer loops beyond the standard three cannot be expected to improve o-a and o-b fits. On the other hand, the reason why these fits are actually degraded is not clear at the moment and is the subject of current investigations.

## 5.2 Forecast skill – full observing system

The forecast skill scores show a high level of consistency with the analysis skill diagnostics. In Figure 12 we present a selection
of tropospheric forecast skill scores relevant for evaluating standard synoptic performance (500 hPa geopotential RMS forecast error, top row), the water cycle (Total Column Water Vapour RMS error, second row) and the wind field (200 and 850 hPa wind vector RMS errors, third and bottom row). All the diagnostics confirm the significant degradation in performance for the



one outer loop experiment and the small but statistically significant improvement of the four and five outer loop experiments with respect to the baseline three outer loop experiment. In the stratosphere (not shown) forecast skill scores again show degraded performance for the one outer loop experiment, while results are mostly neutral or slightly positive for the four and five outer loop experiments. One notable exception is the tropical stratospheric layer from 5 to 1 hPa where the five outer loop

experiment show a statistically significant degradation, again confirming the analysis skill diagnostic results.

### 5.3 Analysis and Forecast skill - linear observation operators

In Sect. 2 of this paper we have showed how nonlinear effects in 4D-Var arise from two different sources: nonlinearities in the model evolution during the assimilation window and nonlinearities in the observation operators. It is difficult to cleanly disentangle the two effects, as they are linked inside the generalised observation operator G and its linearisations. We have

however tried to evaluate the impact of the model nonlinearities in isolation by running a set of multi-outer loop assimilation experiments where we have retained a subset of observations that are linearly related to the control variables (Conventional in-situ observations, Atmospheric Motion Vectors, GPS Radio Occultation bending angles, microwave temperature sounders). A sample of results from this set of experiments is presented in Figure 13, which shows the same set of forecast skill scores shown in Fig. 12 for the experiments with the full observing system. It can be seen hat the impact of going from one to three

outer loops is still very significant. However, the impact of going from three to four outer loops appear to be smaller than in the experiments with the full observing system, and this effect is visible in other forecast skill measures as well (not shown). This suggests that in the current ECMWF 4D-Var it is the presence of nonlinear observations (in particular the all-sky radiances sensitive to cloud and precipitation) that is responsible for the additional benefit of running more than the current three outer loops.

### 20    6 Discussion and Conclusions

In modern atmospheric data assimilation (and, arguably, in most of the other Earth system components as well) nonlinearities play an ever more important role. This is due to the ever-increasing resolution and complexity of the prognostic models, which exhibit instabilities at smaller scales and thus present more nonlinear error growth during the assimilation window, and to the emergence of an array of observations that are nonlinearly related to the control vector variables used in the variational

analyses. Both these trends are expected to continue in the near future, which makes the capacity of the assimilation algorithms to deal effectively with nonlinear effects an increasingly important benchmark.

The ECMWF implementation of 4D-Var relies on a perturbative approach to nonlinearity. Incremental 4D-Var is based on the concept of a purely linear analysis update iterated on ever more accurate first guess trajectories. Diagnostics in both observation space and model space support this interpretation, and show that the capacity to run more than one outer loop is a significant

driver of the overall ECMWF analysis and forecast skill. Results from long data assimilation cycling experiments show that running the current ECMWF 4D-Var with one outer loop only, which is equivalent to making a purely linear analysis update,



would result in very significant deterioration in all analysis and forecast accuracy metrics. Conversely, adding one, or possibly two additional outer loops to the current operational set-up of three outer loop updates, appear beneficial both in terms of analysis quality and in terms of general forecast skill. Results from limited additional experimentation (not shown) also indicate that more than five outer loops do not appear to bring further benefits, at least in the experimental configuration we have used.

One interesting question is about the limits of applicability of the multi-incremental approach in the ECMWF data assimilation system. As noted in Sec. 5, while the tropospheric analyses and forecasts were consistently improved in the four and five outer loop assimilation experiments, signs of degradation started to appear in the analysis and first guess fit of some types of stratospheric peaking radiance observations. Interestingly, these degradations were not seen in the experiments using only observations which are linearly related to the state. This suggests that changes to the analysis introduced by the assimilation

of nonlinear observations (mainly humidity and cloud and precipitation sensitive radiances) affect the stratospheric analysis either through the shape of the background error spatial correlations or by the generation of gravity wave structures in the initial conditions. These interactions are currently being investigated.

Another obvious factor potentially limiting the applicability of the incremental algorithm is the range of validity of the tangent linear (TL) hypothesis (Sect. 2). As reported in Bonavita et al. 2017b, problems in 4D-Var convergence connected with the

TL hypothesis usually arise in situations where the first guess departures are at least one order of magnitude larger than the assumed observation errors. In most cases, use of more realistic values of the observation errors, which better take into account the representativity and observation operator components, are sufficient to regularise the minimization.

While the advantages of being able to run an increased number of outer loop linearisations is clear, the question remains on how to fit them inside the typically tight operational schedules of operational weather centres. Taking the ECMWF data

assimilation system as an example, the three outer loops 4D-Var analysis has about 45 minutes to complete. Given the sequential nature of the 4D-Var minimization, each additional outer loop would increase this time by approx. 15 minutes. This implies that, in the current set-up, the observation cut-off time would have to be pushed back by a similar time interval, quickly negating any advantage that the increased number of outer loops might bring. One possible way to overcome this problem would be to allow late arriving observations to enter the assimilation at successive outer loop updates. This would effectively

push the observation cut-off time forward to the beginning of the last minimization, thus allowing to start the 4D-Var analysis earlier and thus accommodate additional outer loop updates. This assimilation framework, which we call "continuous DA", is currently being tested at ECMWF and results will be documented in a forthcoming paper. Note that in the continuous DA the problem being solved is conceptually different from that of incremental 4D-Var. In incremental 4D-Var we solve a nonlinear problem through repeated linearisations. In the continuous DA we solve a sequence of slightly different nonlinear minimisation

problems, taking advantage of increasingly accurate first guess trajectories.

Another possible approach to increase the number of outer loops within the operational time constraints is to adopt an "overlapping" assimilation window framework, for example along the lines discussed in Bonavita et al, 2017a. In this configuration, observations that have been assimilated in both successive overlapping windows will have effectively be seen



by twice the number of guess trajectories as in a standard non-overlapping configuration. This idea, similar to the quasi-static variational DA approach of Pires et al, 1996 and Jarvinen et al, 1996, is also being actively investigated.

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



| Experiment | Resolution of Outer Loop | Number of Outer Loops | Resolution of Minimizations |
|---|---|---|---|
| Reference | TCo399 | 3 | TL95/TL159/TL255 |
| 1 Outer Loop Experiment | TCo399 | 1 | TL255 |
| 4 Outer Loop Experiment | TCo399 | 4 | TL95/TL159/TL255/TL255 |
| 5 Outer Loop Experiment | TCo399 | 5 | TL95/TL159/TL255/TL255/TL255 |

**Table 1. Resolution and number of outer loop iterations for the sensitivity experiments discussed in Sec. 5. TCo399 means IFS model integrations with spectral triangular truncation 399 and a cubic octahedral reduced Gaussian grid. TLXXX mean IFS model integrations carried out at spectral triangular truncation XXX on a linear reduced Gaussian grid.**

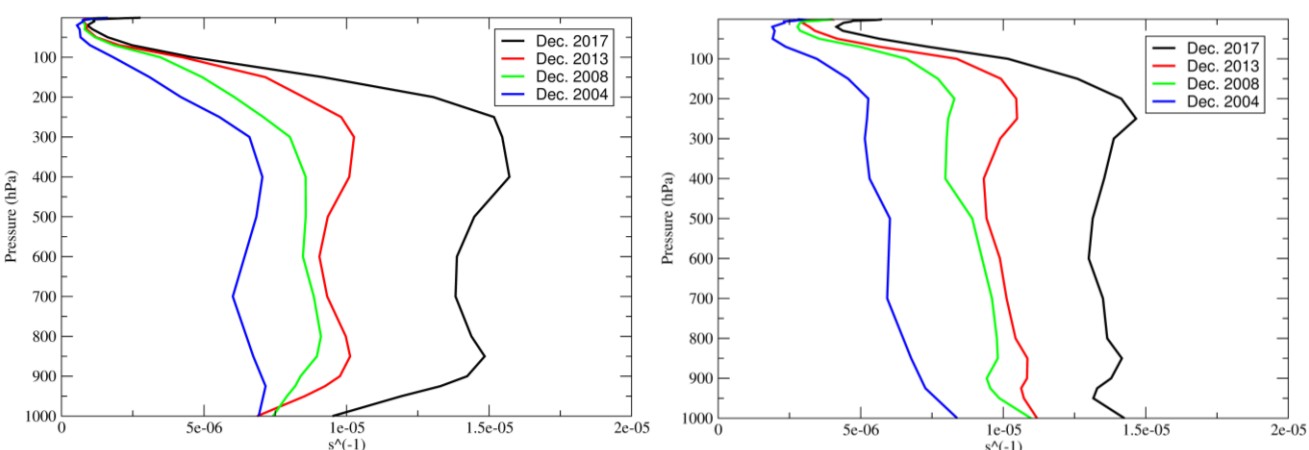

**Figure 1. Globally averaged profiles of historical ECMWF 4D-Var differences $M(x^{n-1} + \delta x^n) - (M(x^{n-1}) + \mathbf{M}\delta x^n)$ in the last minimization 9 hours into the 12h assimilation window for vorticity (left) and divergence (right) in 2004, 2008, 2013 and 2017. Over the years, the resolution and number of inner and outer loops have increased from 60-level TL511/TL95-TL159 in 2004 to 137-level TCo1279/TL255-TL319-TL399 in 2017.**





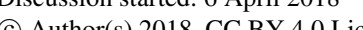

(a) AMSU-A channel 6

(b) Radiosonde temperature

(c) ATMS channel 20

(d) AMSR-2 channel 11

**Figure 2. Ensemble mean $G(x)$ (x-axis) against control member $G(x)$ for (a) AMSU-A channel 6 in clear sky locations (b) radiosonde temperature (c) ATMS channel 20 (clear sky) and (d) ASMR-2 channel 11 observations (all-sky).**

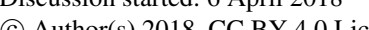



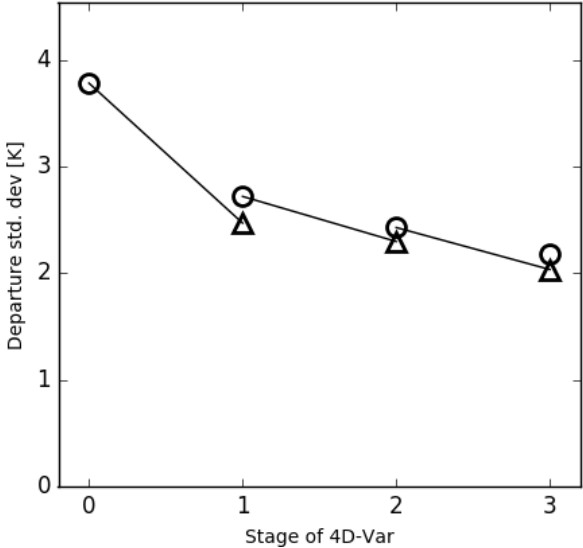

**Figure 3. Standard deviation of departures for AMSR-2 channel 10 in the nonlinear trajectories (circles) and linear minimisation steps (triangles) for each outer loop of 4D-Var. Results from a single cycle from the ECMWF operational assimilation system.**

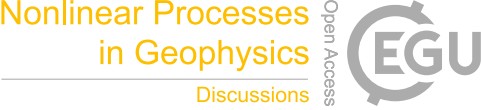



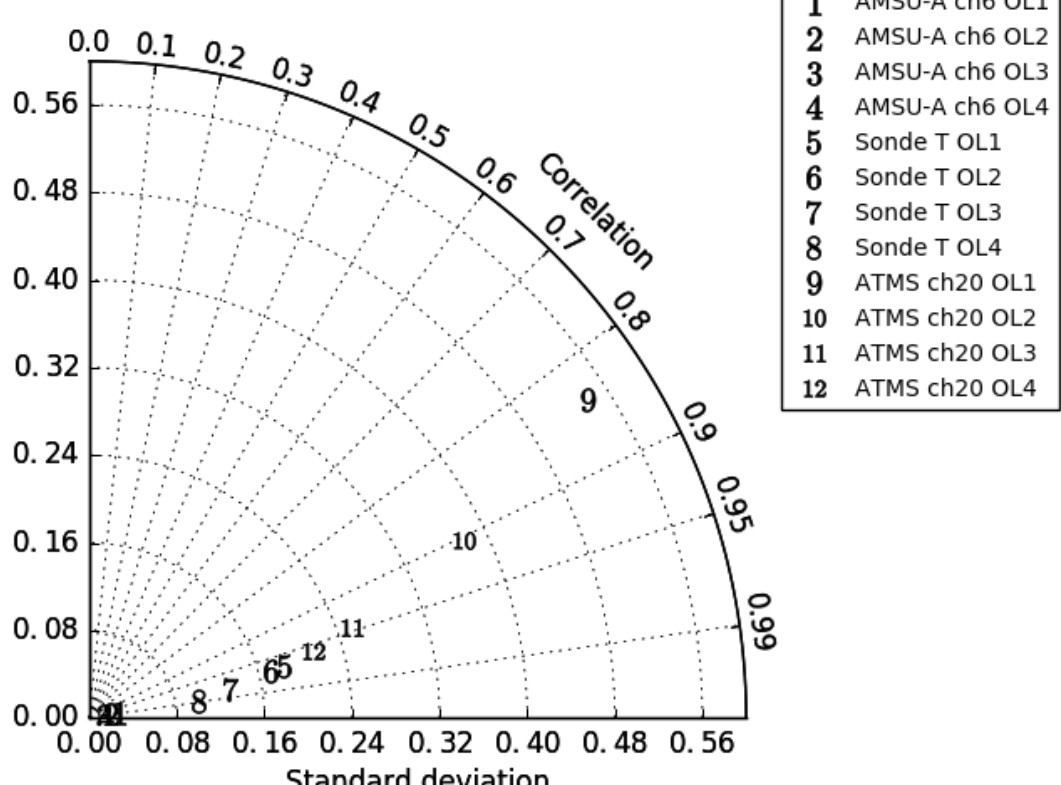

**Figure 4. Taylor diagram showing the correlation (azimuthal angle) and standard deviation (distance from the origin) of differences in observation departures between nonlinear trajectory and linear minimisation steps for each outer loop for different observation types (ASMU-A channel 6, radiosonde temperature and ATMS channel 20).**

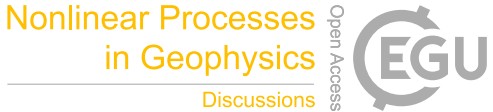

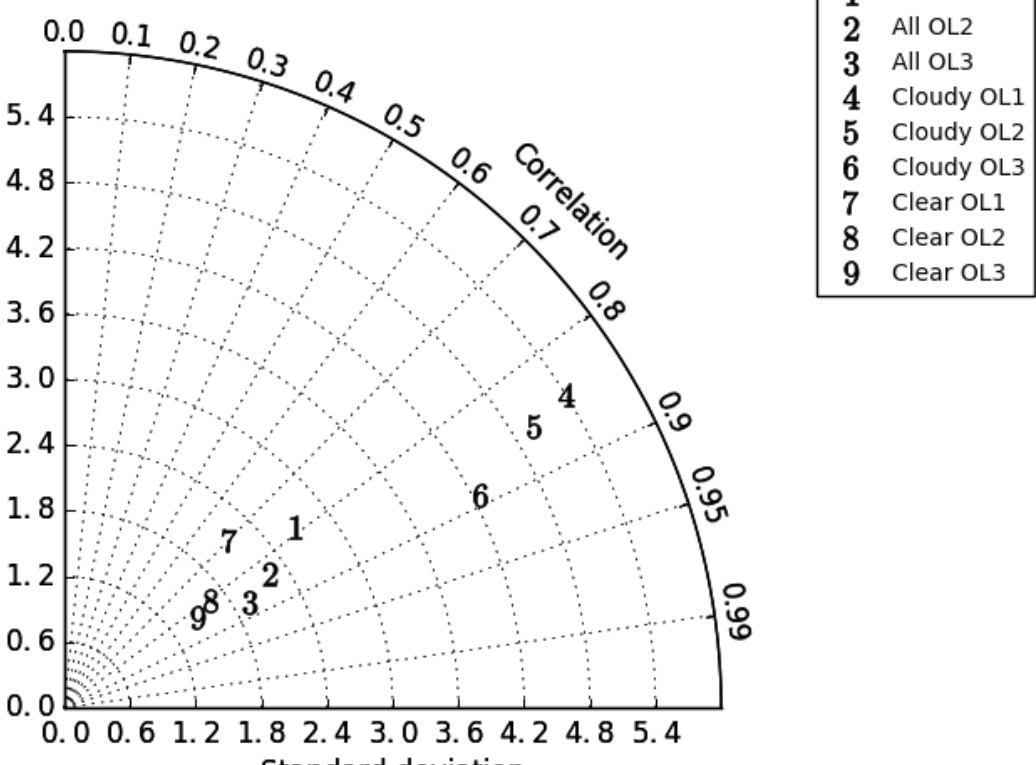

**Figure 5. As Fig. 4 but showing results from AMSR-2 channel 11 categorising observations by those in clear sky regions and those impacted by cloud.**

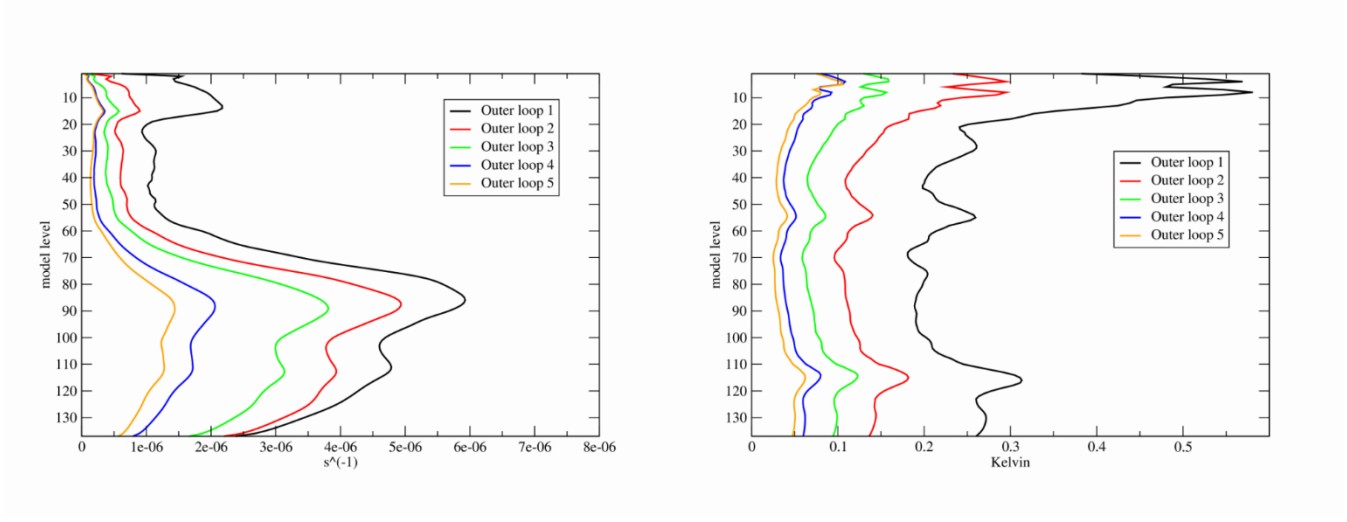



**Figure 6: Vertical profiles of the globally averaged standard deviation of the analysis increments produced by successive outer loop iterations for vorticity (left panel) and temperature (right panel). Values have been averaged over a one-month period. The assimilation experiment has been run with an outer loop resolution corresponding to a cubic octahedral reduced Gaussian grid with spectral truncation 399 (TCo399, approx. 30 km grid spacing); and inner loop resolutions corresponding to linear reduced resolution**

5  **Gaussian grids at spectral truncations TL95/159/255/255/255, corresponding to approx. 210/120/80 km grid spacing. Model level 115 approx. corresponds to the 850 hPa isobaric surface; model level 80 to 250 hPa; model level 15 to 1 hPa.**

**Figure 7. Vertical profiles of the longitudinally averaged standard deviation of the analysis increments produced at the end of the first outer loop minimisation (left column) and the fifth outer loop minimisation (right column) for temperature (first row), vorticity (second row) and humidity (third row). Details of the assimilation experiments as described in Fig. 4.2.1.**



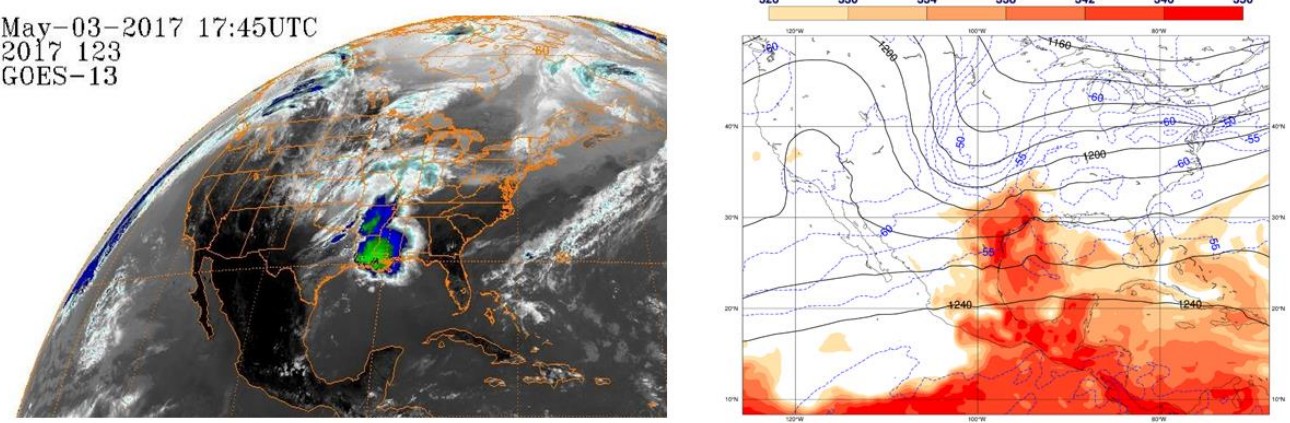

**Figure 8. Left panel: Infrared image of the continental US from the GOES-13 geostationary satellite, valid on 2017-05-03 17.45 UTC (credits: National Centers for Environmental Information, NOAA). Right panel: ECMWF operational analysis of geopotential at 200 hPa (continuous isolines), temperature at 200 hPa (dashed isolines) and equivalent potential temperature at 850 hPa (shaded, units Kelvin), valid on 2017-05-03 18 UTC.**

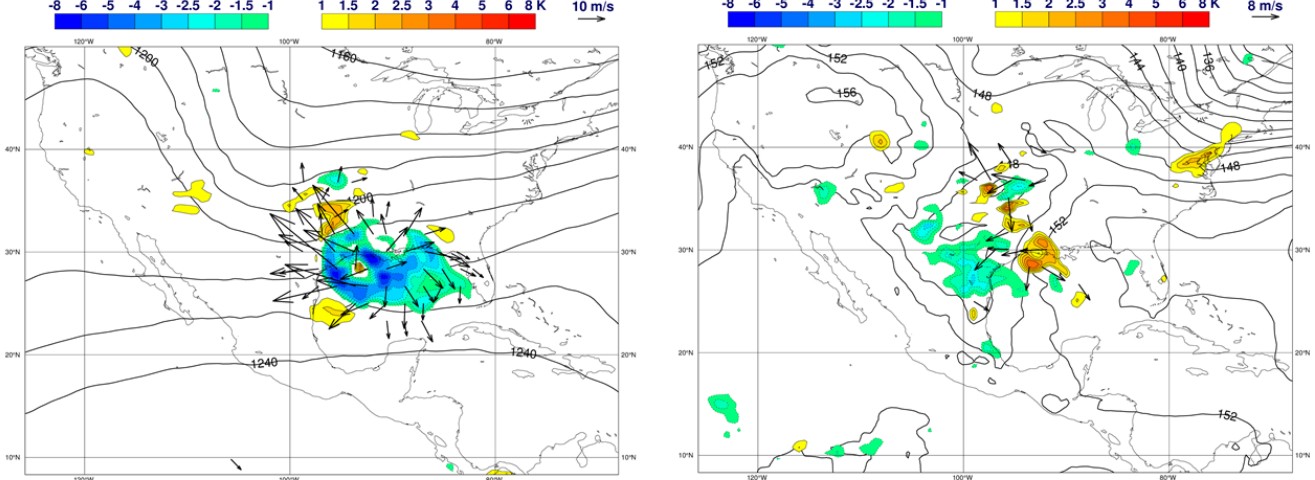

**Figure 9 Left panel: ECMWF operational analysis of geopotential at 200 hPa (continuous isolines), temperature analysis increments at 150 hPa (colour shaded, units Kelvin) and wind vector analysis increments at 150 hPa (arrows, units m/s), valid on 2017-05-03 12 UTC. Right panel: as left, all quantities at 850 hPa.**





**Figure 10. Standard deviation of wind vector observation minus model departures over the 09 to 21 UTC assimilation window of 2017-5-03 in the 100-400 hPa layer for a pre-operational version of the IFS 43R3 cycle: first guess departures (top left panel); a one outer loop analysis departures (top right panel); a three outer loop analysis departures (bottom left plot); a five outer loop analysis departures (bottom right panel).**



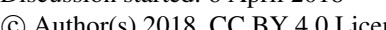

**Figure 11.** Normalised standard deviations of analysis (o-a) and first guess (o-b) departures for radiosonde temperature observations (top left); radiosonde humidity observations (top right); the microwave imager AMSR-2 (bottom left); the combined observations from the AMSU-A microwave sounder instrument on board the AQUA, METOP-A/B, NOAA-15/18/19 satellites (bottom right). The 100% baseline refers to the three outer loop experiment, the black/red/green lines to the one/four/five outer loop experiments respectively. Values smaller/larger than 100 indicated tighter/looser fit of the analysis/first guess to the observations relative to the three outer loop baseline experiment. Values are averaged over the 2016-12-20 to 2017-02-28 period. Error bars represent 95% confidence levels.





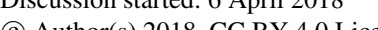

**Figure 12. Normalised root mean square forecast errors for geopotential at 500 hPa (top row); Total Column Water Vapour (second row); Wind Vector at 200 hPa (third row); Wind Vector at 850 hPa (bottom row). The black/red/green lines refer to the one/four/five outer loop experiments, respectively. Errors are normalised with respect to the three outer loop experiments and are computed using the operational ECMWF analysis as verification. Error bars represent 95% confidence levels.**

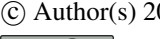



**Figure 13. Normalised root mean square forecast errors for geopotential at 500 hPa (top row); Total Column Water Vapour (second row); Wind Vector at 200 hPa (third row); Wind Vector at 850 hPa (bottom row) for the assimilation experiments using the linear observations subset. The black/red lines refer to the one/four outer loop experiments, respectively. Errors are normalised with respect to the three outer loop experiment and are computed using the operational ECMWF analysis as verification. Error bars represent 95% confidence levels.**