# Peer review of "Nonlinear effects in 4D-Var"

_Nonlinear Processes in Geophysics, 2018_

## Referee Comment (RC1) · Anonymous Referee #1 · 2 Jun 2018

This paper contains a discussion of the effects of nonlinearity in the the ECMWF incremental 4D-Var system. Nonlinearity can affect the 4DVar calculation through the nonlinearity of the dynamical model or through the nonlinear relations between some observed quantities and the model state. As new data sources come on stream, the effects of nonlinear observation functionals become more important than they had been previously. Much of the earlier work on nonlinear effects in data assimilation was done with simplified or schematic numerical models, intended to illustrate particular nonlinear phenomena.

In the "incremental 4DVar" methods used widely in numerical weather prediction and elsewhere in the analysis of the ocean and atmosphere, rather than solving the full variational problem all at once, one assumes that the first guess solution, aka the "background," is a reasonably good estimate of the state of the atmosphere, and we may seek small changes to the background state. The assumption that the changes we

make are small leads to linearization of model dynamics and observation functionals. This cost minimization subject to the assumption that the increment should be small constitutes the "inner loop."

The fact that the assumption of small increments can be problematical in today's NWP environment is complicated further by the fact that the inner loop calculations are usually done at lower resolution than the main model. This means that, in order to perform the inner loop, the main model state must be transferred to the lower-resolution state in which the inner loop calculations are performed, then, when the inner loop is complete, the resulting increment is projected on the high resolution state space and added to the central forecast. This results in a new high resolution analysis, which is used to compute a new central forecast. A new linearization can then be derived to be used in a succeeding inner loop. As time has gone on, the differences in resolution between the central forecast and the inner loop have increased, and thus the model fields can be extremely rough on the coarse resolution inner loop grid due to nonlinearity of the central dynamical model.

The work described in this manuscript represents an attempt to make quantitative sense of the problem of nonlinearity in the ECMWF forecast system. This is a departure from earlier work on simplified systems. It is obviously much more difficult with the full ECMWF forecast system to investigate quantitative and qualitative dynamics of the atmosphere, as is done in earlier studies. The emphasis here is to try to understand the limitations of the incremental approach, and to investigate the separate effects of model nonlinearity and data nonlinearity. I like this paper very much, and strongly recommend publication in essentially its present form. I would like to see a few points cleared up, and a few things done to make the article more accessible to those not directly involved in NWP.

Comments follow:

P4, line 25: "...a nonlinear dependence of the J_B part of the cost function on the

state." The "J_B part of the cost function" is not defined, but I presume, based on common terminology in the literature that the "J_B term" is the first term to the right of the equal sign in (1). That is, of course, quadratic rather than linear. I'm guessing they mean that the nonlinear change of variables leads to a nonlinear dependence of the gradient of the J_B part of the cost function on the state.

P6, line 10 and following, discussion of figure 3: The solid line in figure 3 appears to connect the centers of the circles with the centers of the triangles. This should be noted in the caption. I would be interested to see the prior estimate of the standard deviation, and/or the RMS instrument error, including representation error, marked on that graph. I understand that it might not be so easy to arrive at such numbers.

P6, lines 18-24, discussion of figures 4 and 5: "Figure 4 plots the correlation and standard deviation of these differences ..." What are the units on the radial axes? Are standard deviations for different observed quantities normalized to give a single number for the total standard deviation? If so, how?

P7, line 1: "The magnitude of the analysis increments is seen to gradually decrease ..." Can the authors extract a rate of convergence of the iteration process? Given some reasonable set of assumptions, e.g., local convexity of the nonlinear cost function, do we know what rate of convergence to expect?

It is well known that for the cost function based on linearized assumptions, the value of the cost function at its minimum should be a random variable with chi-square distribution on a number of degrees of freedom equal to the number of independent observations. How close do the cost functions discussed here come to satisfying the chi-square test at, say, 95% confidence?

p8, line 11: "No further improvements were seen with further increases in the outer loop count, which points to residual model deficiencies ..." Another possibility is the observation error, again including representation error. Can it be that the model is already fitting the data within the assumed level of data error?

Figures 6 and 7 need better labels on the ordinate axes. In figure 6, the ordinate axis is labeled "model level." In figure 7 the ordinate axis is not labeled at all. Please change the labels in both to pressure.
* * *

---

## Author Comment (AC1) · 14 Jun 2018

We thank the Reviewer for the careful reading of our manuscript and the constructive suggestions. We provide below detailed answers to her/his comments. An edited version of the manuscript is provided in the attachment.

1) P4, line 25: ". . .a nonlinear dependence of the J_B part of the cost function on the state." The "J_B part of the cost function" is not defined, but I presume, based on common terminology in the literature that the "J_B term" is the first term to the right of the equal sign in (1). That is, of course, quadratic rather than linear. I'm guessing they mean that the nonlinear change of variables leads to a nonlinear dependence of the gradient of the J_B part of the cost function on the state.

The Reviewer is correct. Eq. (3) has been expanded to explicitly indicate the J_B and J_O terms of the cost function. Also, the sentence on P4, line 25 has been reworded

to take the Reviewer's comment into account.

2) P6, line 10 and following, discussion of figure 3: The solid line in figure 3 appears to connect the centers of the circles with the centers of the triangles. This should be noted in the caption. I would be interested to see the prior estimate of the standard deviation, and/or the RMS instrument error, including representation error, marked on that graph. I understand that it might not be so easy to arrive at such numbers.

The caption of Figure 3 has been modified to highlight what the Reviewer noted. Also, the background error standard deviation in observation space for the observation type in the plot has been documented in the caption.

3) P6, lines 18-24, discussion of figures 4 and 5: "Figure 4 plots the correlation and standard deviation of these differences . . ." What are the units on the radial axes? Are standard deviations for different observed quantities normalized to give a single number for the total standard deviation? If so, how?

The units on the radial axes are Kelvin. We have plotted the standard deviation of the differences of the departures between the linear and non-linear stages of 4D-Var. For satellite observations, these are departures in the brightness temperature [K] and for radiosonde observations these are departures in the temperature [K]. We have improved the figure caption to provide a better description of what has been plotted. We have not normalised the standard deviations in this figure.

4) P7, line 1: "The magnitude of the analysis increments is seen to gradually decrease . . ." Can the authors extract a rate of convergence of the iteration process? Given some reasonable set of assumptions, e.g., local convexity of the nonlinear cost function, do we know what rate of convergence to expect?

To the Authors' knowledge there are no practically applicable convergence results for incremental 4D-Var. We could certainly estimate a convergence rate from our experimental results, but we do not think this would be of general interest, as it would be

sensitively dependent on the specifics of the experimental setup, and in particular on the resolution and time step used in the outer and inner loops. Also, as it is apparent from the results shown in the paper, convergence may be achieved for a given number of outer loop iterations in some parts of the control space (e.g., stratosphere) and not in others (e.g., tropical troposphere).

5) It is well known that for the cost function based on linearized assumptions, the value of the cost function at its minimum should be a random variable with chi-square distribution on a number of degrees of freedom equal to the number of independent observations. How close do the cost functions discussed here come to satisfying the chi-square test at, say, 95% confidence?

The Reviewer raises an interesting point. Increasing the number of iterations reduces somewhat the value of the Jo term of the cost function, but not enough to significantly alter the chi_squared statistics. These statistics show that for conventional observations the current ECMWF assimilation system is close to statistical consistency (i.e., the normalised cost function is in the 0.4-0.5 range), while for satellite observations values of the cost function are typically in the 0.1-0.2 range. This is due to the significant spatial and spectral error correlations of these observing systems and thus the reduced number of effective independent pieces of information that these observations provide.

6) p8, line 11: "No further improvements were seen with further increases in the outer loop count, which points to residual model deficiencies . . ." Another possibility is the observation error, again including representation error. Can it be that the model is already fitting the data within the assumed level of data error?

The prescribed wind error standard deviations vary as a function of height and specific observing system, but in the atmospheric layer shown in Figure 10 (100-400 hPa) they are in the range of 2 to 2.5 m/s for both zonal and meridional components. This implies wind vector observation errors of $\sim$3 to 3.5 m/s. As it is evident from Figure 10, even

after 5 outer loops the analysed nonlinear trajectory still presents localised wind vector differences significantly greater than 4m/s. We concur with the Reviewer in that, given the rather extreme weather conditions, this residual discrepancy could be accounted for by representativeness errors of the observations themselves. We have modified the text accordingly to allow for this possibility. One could also argue that if we think of representativeness errors as model errors projected into observation space, then this is just a semantic distinction.

7) Figures 6 and 7 need better labels on the ordinate axes. In figure 6, the ordinate axis is labeled "model level." In figure 7 the ordinate axis is not labeled at all. Please change the labels in both to pressure.

This has been done.

Please also note the supplement to this comment:
https://www.nonlin-processes-geophys-discuss.net/npg-2018-20/npg-2018-20-AC1-supplement.pdf

**Supplement:**

**Nonlinear effects in 4D-Var**

Massimo Bonavita1, Peter Lean1, Elias Holm1

[revised manuscript text omitted]
(\mathbf{x}^{n-1} + \delta \mathbf{x}^n) - (M(\mathbf{x}^{n-1}) + \mathbf{M}\delta \mathbf{x}^n)$$
(6)

- 10 and the globally averaged profile of the standard deviation of this quantity is shown in Fig. 1 for selected years from 2004 to 2017. Over the years, there has been an increase in the resolution of the trajectory and the inner loops and the gap in resolution between the two has increased. This has resulted in increased differences, which we interpret as increased nonlinearity due to the combination of increased model resolution and resolution differences between the inner loop and the trajectory. One way to counteract the nonlinearity that comes with resolution increases is to shorten the length of the assimilation window. This
- 15 can be achieved either by very short windows or by the use of overlapping assimilation windows. In this second case the reduction in nonlinearity is realised by reducing the size of the analysis increments  $\delta x^n$  as each new window will start from a first guess trajectory that has already seen the observations in the overlapped part of the window.

**3.2** The role of the observations**

20

The significance of nonlinearities in the observation operators can be estimated using statistics from the Ensemble of Data Assimilations (EDA, Isaksen et al., 2010) system which is run operationally at ECMWF. Each ensemble member is initialised

using a perturbed model state with perturbations drawn from a distribution with zero mean. For linear observation operators, it follows that the ensemble mean of the model equivalents provided by the observation operators is expected to be close to the unperturbed control member (in fact, it should match it exactly in the limit of infinite ensemble size):

$$\frac{1}{N}\sum_{i=0}^{N}G_k\left(x_0+\delta x_i\right) = \frac{1}{N}\sum_{i=0}^{N}G_k\left(x_0\right) + \frac{1}{N}\sum_{i=0}^{N}G_k\left(\delta x_i\right) = 
[revised manuscript text omitted]

---

## Referee Comment (RC2) · Anonymous Referee #2 · 12 Jul 2018

The nonlinear aspects of the incremental 4D-Var data assimilation system used at ECMWF are examined in this paper. This is done by comparing the TLM integration with two nonlinear trajectories, by examining the variation in the fit to the observations after each outer loop and by looking at the magnitude of the analysis increments after each outer loop. The impact of the number of outer loops used in the analysis on the forecast skill is also assessed. It is shown that the degree of nonlinearly in the ECMWF 4D-Var has significantly increased between 2004 and 2017 because of the increasing complexity and resolution of the forecast model and because a greater number of observations sensitive to humidity and cloud (i.e. all-sky) are now assimilated.

This article is well suited for the journal Nonlinear Processes in Geophysics. I suggest that this paper requires only minor revision subject to the following comments:

P4, line 24: Holm et al., 2003 should be Holm et al. 2002.

P6, section 4: The experimental design is not clearly defined in this section. The details about the multi-incremental strategy, like the number of outer loops, the horizontal model resolution and the type of simplified physics (dry vs full) in each loop should be described. Part of this information is found in the caption of Fig.6 as well as in section 5 for the operational ECMWF 4D-Var. It would be clearer if all these pieces of information were gathered at the beginning of this section.

P7, line 2-3: It is stated that '...the magnitude of the analysis increments appears to be asymptote to a relatively small value...' In principle, the analysis increments should converge to zero after a given number of outer loops. Is there an explanation why the analysis increments seem to converge to an asymptote for the temperature and in the stratosphere?

P8, line 29-31: It is stated that the number of inner loop iterations is approximatively 30 for each outer loop and this number is not sensitive to the resolution and outer loop number when using a convergence criterion based on the information content. It is assumed here that a satisfactory convergence is reached for each outer loop. For the experiment with only one outer loop at TL255, is the minimization also stopped after 30 iterations? In this case it is possible that the convergence for all spatial scales may not be reached, which may explain why the results shown in Fig. 13 are much worse for one outer loop than for three and four outer loops. In effect, a degradation of ∼20% in Z500 for 12h forecast lead time in the southern hemisphere extra-tropics is surprisingly large. Furthermore, this result seems not sensitive to the degree of non-linearity of the observation operations according to Fig. 13 (linear operators only) and Fig. 12 (all operators). I recall here that the main advantage of the multi-incremental strategy proposed by Veerse and Thepaut (1998) is to find the minimum of the 4D-Var cost function at a lower and affordable computational cost by exploiting the fast convergence of large-scale increments at the beginning of the minimization. The full convergence of the minimization problem for the first outer loops is not necessary since the re-linearization and the change in resolution modify the shape of the quadratic cost

function. Most importantly is the convergence of the last outer loop. The convergence of smaller-scale increments is slower and hence embedded in the inner/outer loops process. For a fair comparison, the total number of inner loop iterations should ideally be the same for the experiments with one and three outer loops (∼90 iterations). This is what it is done in Veerse and Thepaut (1998) where 80 iterations are used for both the one outer loop (80) and four outer loops (4x20) experiments. If only about 30 iterations are used in the one outer loop experiment, then it is important to verify that the gradient norm decreases by at least two or three order of magnitude between the first and last iteration. I suggest that a brief discussion about this issue be added to the text.

---

## Referee Comment (RC3) · Anonymous Referee #3 · 13 Jul 2018

This manuscript provides a very detailed study of possible effects of nonlinearities in the ECMWF hybrid 4DVAR system. It revisits the tests that were undertaken when the resolutions of both the nonlinear and the linearised models were a lot lower than today's operational configuration, but also the effects now of all sky radiances with the nonlineary behaviour of moisture and hydrometeors.

This a very well put together study, and I only have a few technical changes to the text and grammar.

1) Please use the notation as set out in Ide et al (1997) Nonlinear observation operators should be denoted by a bold h, the tangent linear by the H, but also that linear observations should be denoted as Hx. This became confusing in equation 7, yet you do use the correct formulation for the tangent linear model of the numerical model in equation 6.

2) Page 2: This paragraph is far too long. Please break it up into smaller chunks. I would suggest on lines 6, 15, 20 and 24.

3) On Line 32 on the same page i would recommend starting the sentence as given the motivation above, the remainder of this paper is .... Also on line 34 i am not sure if the is should be an are as you have the plural of effect in the sentence.

4) There should be punctuation after every equation as they should be considered as part of the sentence.

5) Equation 3 i believe is incorrect in that \delta x_0 is usually defined as x_0-x_b so that the background term is usually just expressed in terms of the increments, If this is not the case you need to define \delta x_0 which is not done here.

I highly recommend that this manuscript be accepted for publication.
* * *

---

## Author Comment (AC2) · 14 Jul 2018

Response to Reviewer of "Nonlinear effects in 4D-Var" by Massimo Bonavita, Peter Lean and Elias Holm.

Anonymous Referee #2 We thank the Reviewer for the careful reading of our manuscript and the constructive suggestions. We provide below detailed answers to her/his comments.

1) P4, line 24: "Holm et al., 2003 should be Holm et al. 2002. ".

Thanks for spotting this typo, it will be corrected in the final version.

2) P6, section 4: "The experimental design is not clearly defined in this section. The details about the multi-incremental strategy, like the number of outer loops, the horizontal model resolution and the type of simplified physics (dry vs full) in each loop

should be described. Part of this information is found in the caption of Fig.6 as well as in section 5 for the operational ECMWF 4D-Var. It would be clearer if all these pieces of information were gathered at the beginning of this section.

The details of the multi-incremental 4D-Var setups are described in Sec 5 and in Table 1. They will be referred to in Sect. 4 as well, as requested by the Reviewer.

3) P7, line 2-3: It is stated that '... the magnitude of the analysis increments appears to be asymptote to a relatively small value ...' In principle, the analysis increments should converge to zero after a given number of outer loops. Is there an explanation why the analysis increments seem to converge to an asymptote for the temperature and in the stratosphere?

The Reviewer has raised an interesting point. The lack of convergence at outer loop level apparent in the stratospheric temperature analysis increments is mirrored in observation space by the increased analysis and background departures visible in the stratospheric-peaking channels of satellite sounding instruments (Fig. 11, bottom right panel). After some further sensitivity studies, the cause of this behaviour has been traced to the different timesteps used in the inner and outer loops. The difference in timestep leads to different speed of propagation of stratospheric gravity waves, which then leads to oscillating behaviour in the minimization. This will be discussed in the revision, but we defer a more complete treatment of this interesting effect to a future paper.

4) P8, line 29-31: It is stated that the number of inner loop iterations is approximatively 30 for each outer loop and this number is not sensitive to the resolution and outer loop number when using a convergence criterion based on the information content. It is assumed here that a satisfactory convergence is reached for each outer loop. For the experiment with only one outer loop at TL255, is the minimization also stopped after 30 iterations? In this case it is possible that the convergence for all spatial scales may not be reached, which may explain why the results shown in Fig. 13 are much

worse for one outer loop than for three and four outer loops. In effect, a degradation of âĹij20% in Z500 for 12h forecast lead time in the southern hemisphere extra-tropics is surprisingly large. Furthermore, this result seems not sensitive to the degree of non-linearity of the observation operations according to Fig. 13 (linear operators only) and Fig. 12 (all operators). I recall here that the main advantage of the multi-incremental strategy proposed by Veerse and Thepaut (1998) is to find the minimum of the 4D-Var cost function at a lower and affordable computational cost by exploiting the fast convergence of large-scale increments at the beginning of the minimization. The full convergence of the minimization problem for the first outer loops is not necessary since the re-linearization and the change in resolution modify the shape of the quadratic cost function. Most importantly is the convergence of the last outer loop. The convergence of smaller-scale increments is slower and hence embedded in the inner/outer loops process. For a fair comparison, the total number of inner loop iterations should ideally be the same for the experiments with one and three outer loops (âĹij90 iterations). This is what it is done in Veerse and Thepaut (1998) where 80 iterations are used for both the one outer loop (80) and four outer loops (4x20) experiments. If only about 30 iterations are used in the one outer loop experiment, then it is important to verify that the gradient norm decreases by at least two or three order of magnitude between the first and last iteration. I suggest that a brief discussion about this issue be added to the text.

As stated on Page 8, Sec. 5, of the manuscript "The number of inner loop iterations in the minimisations is not prescribed, because minimisations stop when a convergence criterion based on the information content of the minimisation is reached. Convergence is usually reached in approx. 30 iterations, and this number has been found not to be sensitive to resolution and number of outer loop relinearizations". The convergence criterium of the ECMWF 4D-Var minimization is based on an information content type of measure as described in Fisher, 2003 ("Estimation of entropy reduction and degrees of freedom for signal for large variational analysis systems", ECMWF Tech. memorandum n. 397) and it is typically reached in about 30 iterations, with little sensitivity to

resolution and number of outer loop relinearisations. It is important to note that this convergence criterium is always met before the hard stop criterium of 50 iterations is reached in the experiments described in the paper. Regarding the convergence properties of the coniugate gradient/Lanczos algorithm for different spatial scales, it is worth noting that the one outer loop 4D-Var runs with a T255 minimization resolution, which is the highest resolution used in the multi-incremental experiments (Table 1), and it runs to convergence (within the criterium described above). We deduce that the reduced skill of the single outer loop experiment is not due to convergence issues or insufficient resolution of the analysis increments, but to its inability to account for nonlinear effects in both the model and the observations. In terms of how degraded the single outer loop analyses and forecasts are, it is well-known that day 1,2 skill scores are sensitive to the verifying truth and its error correlations with the forecast fields. After day 3, when we can reasonably assume that the verifying analysis errors are small in a relative sense with respect to the forecast errors, then the degradation is seen to be in the 3 to 5% range. This is in line with the results of O-B statistics for different instruments (Fig. 11). As a final point, our main objective in the investigation of the impact of additional outer loops is not to compare performance in an "equal cost" scenario, but to approximately quantify how much skill the additional cost of running more outer loops might give. We will add this discussion in the revised version of our manuscript.

---

## Author Comment (AC3) · 14 Jul 2018

We thank the Reviewer for his/her careful review and his appreciation of our work. We accept his/her suggestions for the technical issues raised which will be rectified in the final version of the paper.

---

## Referee Comment (RC4) · Anonymous Referee #4 · 19 Jul 2018

Nonlinear effects in 4D-Var

by Massimo Bonavita, Peter Lean and Elias Holm

Date:        July 16, 2018

**1    Recommendation**

This paper revisits some important issues studied previously with simpler models several years ago. The objective is to assess the extent to which nonlinearities are correctly dealt with by using an incremental approach in which successive linearizations are used to minimize a non-quadratic objective function. The source of nonlinearities are first and foremost the model itself but also the treatment of humidity and even new formulation of the objective function that include non-Gaussian observation error (e.g., the Huber norm). This is an interesting paper that raises these issues and presents results obtained with the state-of-the-art 4D-Var of ECMWF. Although one could think of many other experiments that could be done, what is presented here is very interesting. The paper is well written and the results presented clearly. My recommendation is then that the paper be accepted with minor corrections. However, the authors should consider the specific comments that I think need to be addressed. I consider those to be minor except for comment p8, Fig10: the different color bars used for the three figures make it difficult to draw conclusions.

**2    Specific comments**

p1L15:      Rabier and Courtier (QJRMS 1992) presented a good study to measure the accuracy of the tangent-linear with a "realistic" global baroclinic model, the IFS of the time but without the physical parameterizations. Lacarra and Talagrand (1988) did also conclude that the TL model was a reasonable approximation of the evolution of a perturbation for ~48-h. This is the basis of ECMWF's EPS.

p1L19:      the reference to Pires *et al.* (1996) is missing in the list of references.

p1L24:      "the use of a linear model".

p2L22:      although this is mentioned in the legend, the notation TL511, for example, should be explained: eg., 40 km (TL511, spectral truncation 511 with a linear grid). The same for TCo1279, which was a new one for me.

p4 eq.(5):  the Hessian should be written as   $\mathbf{B}^{-1} + \sum_k \mathbf{G}_k^T \mathbf{R}_k^{-1} \mathbf{G}_k$  . The perturbation $\delta\mathbf{x}_0$ is not part it.

p3L23:      the notation $\mathbf{x}^t$ is usually used for the true state in the literature while here it refers to the trajectory. I suggest that the notation $\mathbf{x}^{tr}$ should be used instead.

p5L4:I can understnd that interpolation of a high resolution field with high variability would create a "noisy" low resolution field. However, such cases require special treatment of the

interpolaitonto be representative (e.g, aggregation instead of geometrical interpolation). Can the authors comment on this?

p5L10:     Rabier and Courtier (QJ 1992) have looked into this with a similar approach. This should be referred to and discussed in relation with the results of Fig.1. The difference

$$\Delta \mathbf{X} = M\left(\mathbf{x}^{n-1} + \delta\mathbf{x}^{n}\right) - M\left(\mathbf{x}^{n-1} - \delta\mathbf{x}^{n}\right)$$

shows how long the linearity assumption holds regardless of how the TLM has been formulated. If $\Delta\mathbf{X}$ is not small, it implies that it is hopeless to be able to find a linear model that would provide a reasonable evolution of the perturbation in the initial conditions.

p5L15:     reusing the observations implies that the accuracy of the background state has changed and the B-matrix should reflect that. The first-guess trajectory is therefore not the background state or is it? My interpretation is that the first-guess is just a starting point of the minimization as are each state used in an outer loop.

p5L23:     this argument holds if the errors are Gaussian but non-Gaussian errors could lead to a non-zero mean ensemble.

p7L5:     convergence needs to consider that the objective function represents the fit of a given realisation of the observations. Convergence to numerical accuracy is meaningless and it is justified to reduce the requested accuracy of the minimization.

p8, Fig10:  it is difficult to say that the departures with respect to wind observations is smaller when more outer loops are used since the scales (color bars) are not the same. In fact, that of the first-guess has a value of 9.97, while for one outer iteration it is 11.91, with three we get 13.52 and finally with 5, 11.91. It is not so "visually" apparent that 5 outer loops is better.

p8L15:     this particular situation involves the physical parameterizations (convection). Looking at the initial physical tendencies (see Rodwell and Palmer, QJ2007) may reveal interesting information for this particular case.

p9L10 (Fig.11):  a reduction of O-A is not a good measure of the quality of the analysis. This can be obtained by reducing the observation error but result in even unphysical forecasts. The O-B is a better measure in that sense and indicates that increasing the outer loop from 3 to 4 or 5 leads to a more modest gain.

With one outer loop, is the resolution of the analysis increment TL95 or does it correspond to that used in the third outer loop (TL399). If it is TL95, the degradation may be attributed to the degraded resolution and in that case, it would be better to redo it using the same higher resolution as used for the 3, 4 and 5 outer loops. Even in 3D-Var, TL95 would be considered too low.

p9L25: with significantly different observation errors, the minimization would focus first on those with small errors and it is only when convergence is reached for those that it would take care of others. This can happen when artificially large observation error are assigned to some satellite observations which were then incapable to have a significant influence on the analysis. In this particular case, it may be that satellite observations have now impacted the analysis significantly more than before.

p10L10:   an experiment assimilating only satellite measurements sensitive to humidity and precipitation could show more about the nonlinearity associated with those.

p10L14:   "seen t̲hat the impact".

p10 Conclusion: increasing the resolution implies a reduction of the nonlinear timescale. The assimilation window would have to be shortened. It is important to evaluate what this timescale is. If the TLM of the full model cannot be achieved, Tanguay et al. (1995) have shown with a simple model that even in the best of cases convergence cannot be reached. To what extent can we say that a weak constraint 4D-Var would be needed? Being at ECMWF hwere the weak constraint 4D-Var has been extensively studied, the authors are in a position to comment on this.

---

## Author Comment (AC4) · 2 Aug 2018

**Response to Reviewer#4 of "Nonlinear effects in 4D-Var" by Massimo Bonavita, Peter Lean and Elias Holm.**

**Anonymous Referee #4**

We thank the Reviewer for her/his careful reading of our manuscript and the constructive suggestions, which we will mostly incorporate into the final version of our manuscript. We also provide below detailed answers to her/his comments.

1) *p1L15: Rabier and Courtier (QJRMS 1992) presented a good study to measure the accuracy of the tangent-linear with a "realistic" global baroclinic model, the IFS of the time but without the physical parameterizations. Lacarra and Talagrand (1988) did also conclude that the TL model was a reasonable approximation of the evolution of a perturbation for ~48-h. This is the basis of ECMWF's EPS.P4, line 24: "Holm et al., 2003 should be Holm et al. 2002. ".*

   Yes, this is an important early work on the validity of the TL approximation for a baroclinic wave evolution, it will be added in the references. Also, thanks for spotting the error in the Holm reference, it will be corrected in the final version.

2) *the reference to Pires et al. (1996) is missing in the list of references.*

   It is present, PG 13 L 27.

3) *p1L24: "the use of a linear model".*

   OK

4) *p2L22: although this is mentioned in the legend, the notation TL511, for example, should be explained: eg., 40 km (TL511, spectral truncation 511 with a linear grid). The same for TCo1279, which was a new one for me.*

   It is a good suggestion, this will be made clearer in the final version. Also, a relevant reference will be added for the reader interested in more details.

5) *p4 eq.(5): the Hessian should be written as $\mathbf{B}^{-1} + \sum_k \mathbf{G}_k^T \mathbf{R}_k^{-1} \mathbf{G}_k$ . The perturbation $\delta\mathbf{x}_0$ is not part of it.*

   Thanks for spotting this error. What we meant to write here is that the Hessian is a function of the reference trajectory in the nonlinear case. This will be rectified in the final version.

6) *p3L23: the notation $x^t$ is usually used for the true state in the literature while here it refers to the trajectory. I suggest that the notation $x^{tr}$ should be used instead.*

   We agree, $x^t$ could be confusing. We will denote the guess trajectory as $x^g$ in the final version of the manuscript.

7) *p5L4: I can understand that interpolation of a high resolution field with high variability would create a "noisy" low resolution field. However, such cases require special treatment of the interpolation to be representative (e.g, aggregation instead of geometrical interpolation). Can the authors comment on this?.*

   This effect has become more visible in recent years due to the increased difference in resolution between inner and outer loops. Work is under way to find a satisfactory solution in term of appropriate time and space interpolation/averaging from the high resolution to the low resolution trajectory.

8) *Rabier and Courtier (QJ 1992) have looked into this with a similar approach. This should be referred to and discussed in relation with the results of Fig.1. The difference*

$$\Delta\mathbf{x} = M(\mathbf{x}^{n-1} + \delta\mathbf{x}^n) - M(\mathbf{x}^{n-1} - \delta\mathbf{x}^n)$$

*shows how long the linearity assumption holds regardless of how the TLM has been formulated. If $\Delta\mathbf{x}$ is not small, it implies that it is hopeless to be able to find a linear model that would provide a reasonable evolution of the perturbation in the initial conditions.*

Yes, this paper will be referred to in relation with the results of Fig. 1.
The additional diagnostic quantity mentioned by the Referee is useful to evaluate the assimilation window length over which the linearity assumption holds regardless of the accuracy of the TL model. From our perspective in this work, the stronger constraint implied by Eq. (6) in the paper is the relevant quantity to look at to evaluate the goodness of the TL approximation for a fixed assimilation window length and the available TL model.

9) *p5L15: reusing the observations implies that the accuracy of the background state has changed and the B-matrix should reflect that. The first-guess trajectory is therefore not the background state or is it? My interpretation is that the first-guess is just a starting point of the minimization as are each state used in an outer loop.*

The interpretation of the Referee is correct. The guess trajectory is just a reference state around which the generalised observation operator **G** is linearised. Thus, no change in the B-matrix is required.

10) *p5L23: this argument holds if the errors are Gaussian but non-Gaussian errors could lead to a non-zero mean ensemble.*

True. This will be noted in the final version of the paper.

11) *p7L5: convergence needs to consider that the objective function represents the fit of a given realisation of the observations. Convergence to numerical accuracy is meaningless and it is justified to reduce the requested accuracy of the minimization.*

A similar point has been raised in the paper's discussion by an earlier Referee and has been addressed there (and will also be in the final version of the paper).

12) *p8, Fig10: it is difficult to say that the departures with respect to wind observations is smaller when more outer loops are used since the scales (color bars) are not the same. In fact, that of the first-guess has a value of 9.97, while for one outer iteration it is 11.91, with three we get 13.52 and finally with 5, 11.91. It is not so "visually" apparent that 5 outer loops is better.*

The scales used are the same except for the last interval, which is different in the four panels in order to accommodate the different maximum values. Apart from the visual impression, the increased analysis fit with increased number of outer loops is demonstrated by the smaller area-averaged values of the standard deviation of the analysis departures (as reported in the text and in the top captions of each panel).

13) *p8L15: this particular situation involves the physical parameterizations (convection). Looking at the initial physical tendencies (see Rodwell and Palmer, QJ2007) may reveal interesting information for this particular case.*

This is a good suggestion, thanks. We will pursue this idea in future work.

14) *p9L10 (Fig.11): a reduction of O-A is not a good measure of the quality of the analysis. This can be obtained by reducing the observation error but result in even unphysical forecasts. The O-B is a better measure in that sense and indicates that increasing the outer loop from 3 to 4 or 5 leads to a more modest gain.*

We beg to differ on this point. In the ECMWF 4D-Var, the O-A departures are computed through a nonlinear model integration started by the analysis fields at the beginning of the assimilation window. Thus, they represent the best fit of a full model trajectory to the observations over the whole assimilation window. Improving this fit without changing the input error statistics implies a better analysis. A different, but closely related question, is how much of this signal can the full model propagate in time to the next assimilation window. The O-B statistics give a measure of this second aspect. The fact that O-A and O-B statistics show changes which are qualitatively similar but quantitatively different are seen as confirmation that the analysis is behaving properly.

15) *With one outer loop, is the resolution of the analysis increment TL95 or does it correspond to that used in the third outer loop (TL399). If it is TL95, the degradation may be attributed to the degraded resolution and in that case, it would be better to redo it using the same higher resolution as used for the 3, 4 and 5 outer loops. Even in 3D-Var, TL95 would be considered too low.*

This point has been raised in the paper's discussion by an earlier Referee and has been addressed there (and will also be in the final version of the paper). In short, the single resolution experiments have been run at the maximum resolution of the multi outer loop experiments (in this case TL255), so the relatively poor performance of the single resolution experiments cannot be attributed to this.

16) *p9L25: with significantly different observation errors, the minimization would focus first on those with small errors and it is only when convergence is reached for those that it would take care of others. This can happen when artificially large observation error are assigned to some satellite observations which were then incapable to have a significant influence on the analysis. In this particular case, it may be that satellite observations have now impacted the analysis significantly more than before.*

We have conducted further experiments on this aspect and we will shortly report on their results in the final version of the paper. In summary, the problematic convergence of the stratospheric-peaking channels is due to the difference in timestep between inner and outer loops, which causes different wave propagation speed in the outer loop trajectories and in the inner loop minimisations. Running with same timestep in both inner and outer loop solves the problem.

17) *p10L10: an experiment assimilating only satellite measurements sensitive to humidity and precipitation could show more about the nonlinearity associated with those.*

Yes, this is a useful suggestion.

18) *p10L14: "seen that the impact".*

Thanks for spotting this typo.

19) *Conclusion: increasing the resolution implies a reduction of the nonlinear timescale. The assimilation window would have to be shortened. It is important to evaluate what this timescale is. If the TLM of the full model cannot be achieved, Tanguay et al. (1995) have shown with a simple model that even in the best of cases convergence cannot be reached. To what extent can we say that a weak constraint 4D-Var would be needed? Being at ECMWF hwere the weak constraint 4D-Var has been extensively studied, the authors are in a position to comment on this.*

Generalised weak-constraint 4D-Var with a time-varying model error term would be a solution to the increased nonlinear effects in 4D-Var, as the Reviewer suggests. However, it comes with its own problems. A fundamental issue is the evaluation of accurate first and second moments of the model error: it is not clear how to do this, at this stage. Another, connected problem, is that this type of evolving model errors will be auto-correlated in time and cross-correlated with the estimated background errors. Thus, significantly more complex forms of the 4D-Var cost function would have to be implemented.

From our perspective, and as stated in our conclusions, the more promising path towards controlling nonlinearity in 4D-Var is through repeated re-linearisations in a DA framework where the length of the assimilation window is progressively increased. This is not a new idea (e.g., Pires et al., 1996; Jarvinen et al., 1996), but we think it is an idea whose time has come to be put into operational practice.

---

## Author Response (AR1)

**Response to Reviewers Comments on "Nonlinear effects in 4D-Var" by Massimo Bonavita, Peter Lean and Elias Holm.**

We thank all Reviewers for their careful reading of our manuscript and their insightful and constructive suggestions, which we have largely incorporated into the final version of our manuscript. We believe this process has significantly improved the quality of the paper.

[revised manuscript text omitted]

---

## Author Response (AR2)

**Response to Reviewers and Editor Comments on "Nonlinear effects in 4D-Var" by Massimo Bonavita, Peter Lean and Elias Holm.**

We thank the Editor and all Reviewers for their careful reading of our manuscript and their insightful and constructive suggestions, which we have largely incorporated into the final version of our manuscript. We believe this process has significantly improved the quality of the paper.

Detailed responses to the Reviewers' comments have been provided during the discussion phase.

With regard to the Editor's final comments, relevant references to the work of Anna Trevisan and her group have been provided and an Acknowledgement section added.

**Nonlinear effects in 4D-Var**

Massimo Bonavita[1], Peter Lean[1], Elias Holm[1]

[1]European Centre for Medium-Range Weather Forecasts, Shinfield Park, Reading, RG2 9AX, UK

*Correspondence to*: Massimo Bonavita (massimo.bonavita@ecmwf.int)

5   **Abstract.** The ability of a data assimilation system to deal effectively with nonlinearities arising from the prognostic model or the relationship between the control variables and the available observations has received a lot of attention in theoretical studies based on very simplified test models. Less work has been done to quantify the importance of nonlinearities in operational, state-of-the-art global data assimilation systems. In this paper we analyse the nonlinear effects present in ECMWF 4D-Var and evaluate the ability of the incremental formulation to solve the nonlinear assimilation problem in a realistic NWP environment.

10  We find that nonlinearities have increased over the years due to a combination of increased model resolution and the ever-growing importance of observations that are nonlinearly related to the state. Incremental 4D-Var is well suited for dealing with these nonlinear effects, but at the cost of increasing the number of outer loop relinearisations. We then discuss strategies for accommodating the increasing number of sequential outer loops in the tight schedules of operational global NWP.

**1 Introduction**

15  The importance of nonlinear effects has been recognised since the early days of the development of 4D-Var (e.g., Gauthier, 1992; Rabier and Courtier, 1992; Miller et al., 1994; Pires et al., 1996). The presence of nonlinearities in either the model or the observations can potentially cause significant deviations from the usual Gaussian distribution assumed to describe observation and background errors in the definition of the 4D-Var cost function. This in turn translates into a more complex topology of the cost function and the potential for multiple minima (e.g., Pires et al., 1996; Hoteit, 2008). In these conditions,

20  finding the global minimum of the 4D-Var cost function for realistic numerical weather prediction (NWP) applications becomes computationally unaffordable and, even if it were possible, the interpretation and usefulness of the result in the case of multi-modal error distributions becomes unclear in a deterministic analysis context (Lorenc and Payne, 2007).

In order to make the variational problem computationally tractable and mathematically well-posed, simplifications are required. One idea would be to reduce the dimensionality of the control vector used in the minization, for example limiting it

25  to the subspace where dynamical instabilities develop during data assimilation cycle (Trevisan and Uboldi, 2004; Carrassi et al, 2008; Trevisan et al, 2010). Another approach starts from recognising that 
[revised manuscript text omitted]
{1}{2}\left(\delta\mathbf{x}_0 + \mathbf{x}_0^{tg} - \mathbf{x}_b\right)^{\mathrm{T}}\mathbf{B}^{-1}\left(\delta\mathbf{x}_0 + \mathbf{x}_0^{tg} - \mathbf{x}_b\right) + \frac{1}{2}\sum_{k=0}^{K}\left(\mathbf{d}_k - \mathbf{G}_k(\delta\mathbf{x}_0)\right)^{\mathrm{T}}\mathbf{R}_k^{-1}\left(\mathbf{d}_k - \mathbf{G}_k(\delta\mathbf{x}_0)\right) = J_B(\delta\mathbf{x}_0) + J_O(\delta\mathbf{x}_0)$$

(3)

In Eq. (3) $\mathbf{d}_k = \mathbf{y}_k - G_k\left(\mathbf{x}_0^{tg}\right)$ are the observation departures around the latest model trajectory and $\mathbf{G}_k = \mathbf{H}_k\mathbf{M}_{t_0 \to t_k}$ is the linearisation of the generalised observation operator around the defined trajectory.

5 In the observation part of the cost function, the so-called "tangent linear (TL) approximation" has been made in going from (1) to (3):

$$\mathbf{y}_k - G_k(\mathbf{x}_0) = \mathbf{y}_k - G_k\left(\mathbf{x}_0^{tg} + \delta\mathbf{x}_0\right) =$$

$$= \mathbf{y}_k - G_k\left(\mathbf{x}_0^{tg}\right) - \mathbf{G}_k(\delta\mathbf{x}_0) - \frac{1}{2}\,(\delta\mathbf{x}_0)^{\mathrm{T}}\left(\frac{\partial\mathbf{G}_k}{\partial\mathbf{x}}\right)_{\mathbf{x}^{tg}}(\delta\mathbf{x}_0) - O(\|\delta\mathbf{x}_0\|^3) \approx \mathbf{y}_k - G_k\left(\mathbf{x}_0^{tg}\right) - \mathbf{G}_k(\delta\mathbf{x}_0)$$  (4)

In the Taylor expansion in Eq. (4), terms of $O(\|\delta\mathbf{x}_0\|^2)$ and higher are neglected (note that if (4) is exactly satisfied, then Eq.
10 (3) is equivalent to Eq. (1)). This approximation, as first noted in Lawless et al, 2005, is equivalent to the standard approximation used in the Gauss-Newton optimization algorithm, i.e. neglecting the second order derivatives of $G_k$ in the Hessian of the cost function:

$$\nabla^2 J = \mathbf{B}^{-1} + \sum_{k=0}^{K}(\mathbf{G}_k(\delta\mathbf{x}_0)\mathbf{G}_k)^{\mathrm{T}}\mathbf{R}_k^{-1}(\mathbf{G}_k(\delta\mathbf{x}_0))(\mathbf{G}_k) - \sum_{k=0}^{K}\left(\frac{\partial\mathbf{G}_k}{\partial\mathbf{x}}\right)_{\mathbf{x}^{tg}}\mathbf{R}_k^{-1}\left(\mathbf{y}_k - G_k\left(\mathbf{x}_0^g\mathbf{x}_0\right)\right) \approx$$

$$\mathbf{B}^{-1} + \sum_{k=0}^{K}(\mathbf{G}_k)^{\mathrm{T}}\mathbf{R}_k^{-1}(\mathbf{G}_k)\sum_{k=0}^{K}(\mathbf{G}_k(\delta\mathbf{x}_0))^{\mathrm{T}}\mathbf{R}_k^{-1}(\mathbf{G}_k(\delta\mathbf{x}_0))$$         (5)

[revised manuscript text omitted]

**Acknowledgements**

One of the Authors (Massimo Bonavita) would like to express his gratitude to the Organisers of the Anna Trevisan Symposium, and to Alberto Carrassi in particular, for setting up a very successful and interesting meeting and for being such attentive and considerate hosts.

The Authors would also like to thank the Reviewers and the Editor for their careful examination of our manuscript and the many constructive proposals for its improvement.

[revised manuscript text omitted]